# Fire Occurrence in Hemi-Boreal Forests: Exploring Natural and Cultural Scots Pine Fire Regimes Using Dendrochronology in Lithuania

**Michael Manton** [1,*], **Charles Ruffner** [2], **Gintautas Kibirkštis** [3], **Gediminas Brazaitis** [1], **Vitas Marozas** [1], **Rūtilė Pukienė** [3], **Ekaterina Makrickiene** [1] **and Per Angelstam** [4,5]

1    Faculty of Forest Sciences and Ecology, Vytautas Magnus University, Studentu 13, LT-53362 Akademija, Lithuania; gediminas.brazaitis@vdu.lt (G.B.); vitas.marozas@vdu.lt (V.M.); ekaterina.makrickiene@vdu.lt (E.M.)
2    School of Forestry and Horticulture, Southern Illinois University, Carbondale, IL 62901-4411, USA; ruffner@siu.edu
3    Nature Research Centre, Akademijos str. 2, LT-08412 Vilnius, Lithuania; gintautas.kibirkstis@gamtc.lt (G.K.); rutile.pukiene@gamtc.lt (R.P.)
4    Department of Forestry and Wildlife Management, Faculty of Applied Ecology, Agricultural Sciences and Biotechnology, Inland Norway University of Applied Sciences, Campus Evenstad, N-2480 Koppang, Norway; per.angelstam@slu.se
5    School for Forest Management, Faculty of Forest Sciences, Swedish University of Agricultural Sciences, SE-739 21 Skinnskatteberg, Sweden
*    Correspondence: michael.manton@vdu.lt

**Abstract:** Fire is an important natural disturbance and a driver of hemi-boreal forest successional trajectories, structural complexity, and biodiversity. Understanding the historic fire regime is an important step towards sustainable forest management. Focusing on Lithuania's hemi-boreal forests, we first mapped the potential natural fire regimes based on the relationship between site conditions, vegetation, and fire frequency using the ASIO model. The ASIO model revealed that all the fire frequency categories (Absent, Seldom, Intermittent, Often) are found in Lithuania. Scots pine forests dominated the often fire frequency category (92%). Secondly, focusing on a fire-prone forest landscape, Dzūkija, we analyzed the fire occurrence of Scots pine forest types using dendrochronological records. We sampled and cross-dated 132 Scots pine samples with fire scars from four dry forest stands (n = 92) and four peatland forest stands (n = 40), respectively. In total, the fire history analysis revealed 455 fire scars and 213 fire events during the period of 1742–2019. The Weibull median fire intervals were 2.7 years (range 1–34) for the dry forest types and 6.3 years (range 1–27) for the peatland forest types. Analysis pre- and post-1950 showed the Weibull median fire interval increased from 2.2 to 7.2 for the dry forest types but decreased from 6.2 to 5.2. for the peatland forest types. A superposed epoch analysis revealed significant precipitation fluxes prior to the fire events after 1950. Thus, the Dzūkija landscape of Lithuania has been strongly shaped by both human and naturally induced fires. The combination of theory (the ASIO model) with the examination of biological archives can be used to help guide sustainable forest management to emulate forest disturbances related to fire. As traditional forest management focusing on wood production has eliminated fire, and effectively simplified forest ecosystems, we recommend introducing educational programs to communicate the benefits and history of forest fires as well as adaptive management trials that use low-intensity prescribed burning of Scots pine stands.

**Keywords:** biodiversity conservation; cultural burning; forest dynamics; forest management; pattern and process; forest disturbance and succession



## 1. Introduction

Natural disturbances in boreal forests range from small to large-scale, have different frequencies and severities, and include both abiotic and biotic mechanisms [1,2]. Boreal for-

est disturbance patterns and successional pathways vary substantially according to regional climate, edaphic conditions, and the life history traits of tree species [3–5]. Historically, across much of Europe, fires from both natural (e.g., lightning) and human ignitions have influenced local forest stand and landscape dynamics and maintained multiple vegetation types for millennia [6–11]. Understanding the historic disturbance regimes can provide crucial lessons towards achieving multifunctional forest landscapes and the maintenance of naturally occurring habitats and species [12–14]. This insight has triggered the emergence of different typologies for natural forest disturbance regimes [15–17]. These models commonly identify three main forest dynamic types: (1) multi-cohort dynamics related to partial disturbances, such as low-intensity surface fires; (2) even-aged succession after severe stand-replacing disturbances, such as crown fires and large blowdowns; and (3) gap dynamics caused by the death of individual trees or small groups of trees in the absence of fire [15,16,18]. In all forest types, considerable amounts of residual pre-disturbance structures often remain [12].

However, human expansion across northern Europe has transformed naturally dynamic forest landscapes into cultural landscapes that often no longer possess natural patterns and processes characteristic of naturally dynamic systems [4,11,19]. Historic land use of hemi-boreal forest ecosystems has had three main outcomes. Firstly, the most productive deciduous forests sites have been converted into agricultural lands [20]. Secondly, natural mixed forests have been transformed into intensively managed forests for sustained wood production with strongly altered disturbance regimes, successional trajectories, homogenized structure, and even-aged distributions that commonly no longer contain old-aged forest stands [19,21–23]. Finally, the least productive forest sites, largely dominated by Scots pine (*Pinus sylvestris*), have not been deforested but nevertheless have become highly modified, even-aged forest stands that have highly effective fire suppression policies [3,8].

The simplification of forest ecosystem patterns and processes has caused losses of naturally occurring species assemblages across many site types and regions [24–27]. Additionally, the focus on maximized sustained yield of wood and biomass has replaced both traditional cultural practices, such as broadcast burning, and natural forest dynamics [8,28]. Fire is only one example of an ecological process that has been effectively minimized throughout the European boreal forests [3,24,28,29].

There are two species groups particularly adapted to fire: (1) fire-dependent species that require fire for the establishment of particular habitats, and (2) species that show a high preference for fire itself [1]. Low-intensity, surface fire is an important characteristic of boreal forests and is often referred to as a resetting agent of forest succession and thus a driver of habitat mosaic maintenance at the stand and landscape scales [30,31]. Four habitat types for the hemi-boreal zone in the EU Habitats Directory that rely on periodic surface fires to maintain species composition and stand structure are the Western Taiga (9010*), the Central European lichen Scots pine forests (91T0), both found on infertile dry soils, Coniferous forests on or connected to, glaciofluvial eskers (9060) and Bog woodlands (91D0*), found on wet, infertile peatlands [32,33]. However, due to efficient fire suppression, coupled with sustained yield forestry practices since the 1950s, these habitat types have become extremely rare throughout much of northern and central Europe [32]. Accordingly, the recent EU Forest Strategy [34] calls for the development of thresholds and indicators for sustainable forest management concerning forest ecosystem conditions and the implementation of "close to nature" forestry towards restoring natural processes and patterns, as well as adapting forests to deal with climate change. Sustainable forest management is defined as the "stewardship and use of forest lands in a way, and at a rate, that maintains their biodiversity, productivity, regenerative capacity, vitality and their potential to fulfil, now and in the future, relevant ecological, economic and social functions" [34].

Therefore, understanding the historic range of variability of disturbance regimes across forested regions is key towards defining conservation benchmarks and implementing sustainable forest management policy [8,24,28,35–37]. Research on land use history and fire regimes forms an increasingly important avenue for generating knowledge and developing

forest management strategies that consider the successional trajectories of current forest networks towards maintaining ecological processes and ecosystem functions [11,38,39], as well as building resiliency against climate change [19,40]. To achieve sustainable forest management goals, natural disturbance dynamics of forest stands and landscape patterns must be understood, and designs made to mimic the patterns and processes of natural forest ecosystems [12,34]. This includes the investigation of key disturbance features, such as the characteristics of soil moisture and fertility, natural disturbances and historic patterns of fire use, changing return intervals, as well as possible climate effects [3,11,41].

In Lithuania, knowledge of both the historic occurrence and cultural influences of fire that have influenced today's hemi-boreal forest landscapes is limited [42,43]. Therefore, continued efforts are needed to reveal the links between past land use and current forest ecosystem processes and patterns [34,44]. The aims of this study are to fulfil these gaps by (i) modelling the natural forest fire regimes based on soil fertility and moisture and vegetation; (ii) performing a localized dendrochronology study of fire history and climate for the most fire-prone area within Lithuania; and (iii) discussing the influence of cultural traditions and historic events on forest fires in Lithuania. Finally, we propose the prudent use of evidence-based knowledge to stimulate the ongoing debates across the Baltic Sea Region on prescribed burning as a management tool to maintain biodiversity and influence successional dynamics for a multitude of desired future conditions spanning recreational, social, timber, water quality, and wildlife resource concerns.

## 2. Materials and Methods

### 2.1. Framework and Study Area

This study used a nested regional and local study area approach [19] and used the ASIO model [16] and dendrochronology to calculate different fire frequencies based on local and regional site conditions. An overview of the main components of this study is presented in Figure 1.

| Step 1. Apply the ASIO forest fire model to Lithuania's forests | Step 2. Scots pine dendrochronology study of fire history and climate | | | Step 3.Discuss cultural and natural influences on fire history towards emulating fire in Scots pine forests |
|---|---|---|---|---|
| | Step 2a. Reconstruct fire histories for Dzukija (DNP & CSNR peatland) | Step 2b. Analyze precipitation in the lead up with key fire events | Step 2c. Correlate historic climate variables with radial growth patterns | |

**Figure 1.** The three-step analytic approach applied in this study to analyze the historic fire occurrence in Lithuania's hemi-boreal forests and explore the natural and cultural influence of Scots pine multi-cohort forests.

In step 1, we apply the ASIO model to understand the potential occurrence of different natural fire regimes in Lithuania. We focused on all of Lithuania (65,300 km$^2$), situated in the hemi-boreal ecoregion [45], which is a transitional zone between the boreal and temperate forest of nemoral Europe [46] (Figure 2, right). Lithuania's forests are characterized by boreal coniferous tree species on poor soil sites and temperate broadleaved tree species on rich soil sites. The Lithuanian landscape is relatively flat and undulating (ranging from 0–293 m a.s.l.). The structure and composition of Lithuanian hemi-boreal forests are

maintained by a combination of natural and cultural disturbances [4,19,47]. Currently, 22,200 km² (34%) of Lithuania is covered with forest (Figure 2, left), of which Scots pine is the dominant stand species (34.5%), followed by silver birch (*Betula pendula*) 22%, Norway spruce (*Picea abies*) 21%, alder (*Alnus glutinosa* and *A. incana*) 13.7%, aspen (*Populus tremula*) 4.7%, oak (*Quercus robur*) 2.3%, ash (*Fraxinus excelsior*) 0.6%, and other species 1.1% [48].

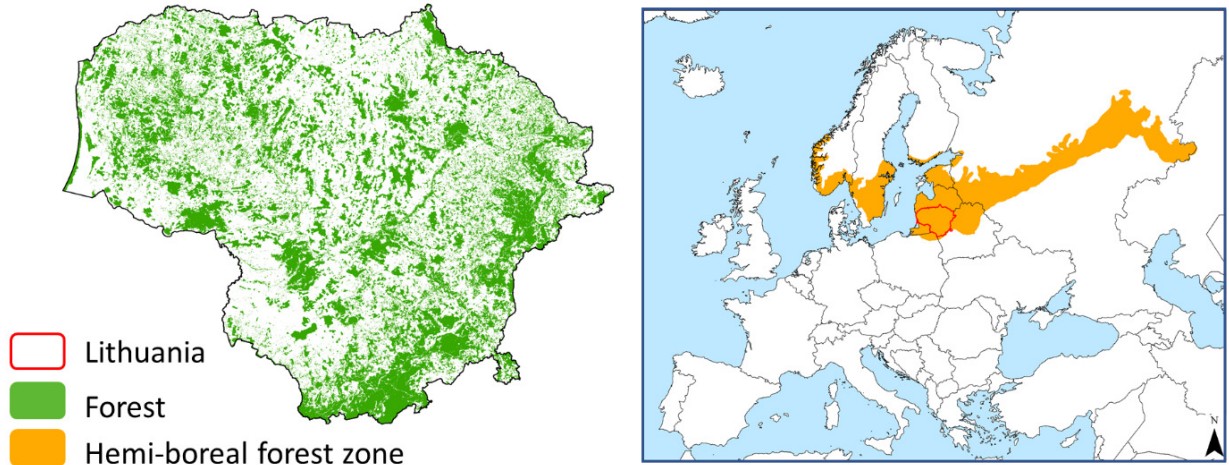

**Figure 2.** Map of Lithuania's forest cover used for the ASIO model analysis (**left**) and the position of Lithuania within the European hemi-boreal forest zone (**right**).

In step 2, we used the predictive ASIO model to identify areas with a high potential for recording surface fires to perform a fire history and dendrochronology analysis. One promising area was the Dzūkija forest landscape with two contrasting Scots pine forest site types: (i) infertile dry sandy soils which are the most fire-prone forest type in Lithuania, based on the ASIO model and (ii) a large peatland complex set within the dry Dzūkija forest landscape (Figure 3). The study area contains the Dzūkija National Park (DNP) (585 km²) and the adjacent Čepkeliai Strict Nature Reserve (CSNR) (112 km²). The DNP was established in 1991 and is the largest national park in Lithuania. Sandy glacial soils dominate with several large peatlands nearby. The CSNR contains the largest peatland complex (58 km²) within Lithuania [49]. It should be noted that Lithuania's national park system is not equivalent to that of the traditional national parks concept in the USA, Canada, Australia, or Sweden [19,50]. For instance, within the Dzūkija National Park, only 4% of the forest area is under strict protection, with the remainder subject to commercial forest harvesting [48]. In contrast, the Čepkeliai Strict Nature Reserve is 100% protected, with restricted access.

In step 2a, we sample, record, and describe the Scots pine fire history for both the dry infertile forests and the wet peatland forests. In step 2b, we analyzed the precipitation fluxes prior to the key fire years using super-posed epoch analysis. In step 2c, we correlated radial growth patterns with historic climatic variables to determine the influence of climate on tree-ring growth.

Finally, in step 3, we discuss the impacts of both the natural and the cultural influences on the fire frequency patterns and propose how low-intensity prescribed fire could be introduced into forest management to enhance forest ecosystems reliant on fire.

### 2.2. Applying the ASIO Model to Lithuania

The ASIO model was originally developed to illustrate that site type characteristics and fire behavior are drivers of forest disturbance regimes. The ASIO model was first applied by the Swedish State Forest Company to guide forest management towards enhancing biodiversity at a landscape scale [51]. The acronym ASIO is based on four levels of fire frequency that are recognized in forests under natural disturbance dynamics [52]. These are: **A**bsent—permanently wet soils where natural fire occurrence is rare (ca. 300 year fire

cycles) [53,54]; **S**eldom—moist sites types that have low fire frequency (ca. 100–150 year fire cycles) [6]; **I**ntermediate—mesic sites types with medium fire frequency (ca. 80–100 year fire cycles) [6]; and **O**ften—dry and infertile soils with a fire frequency of approximately 40–60 years [6,55].

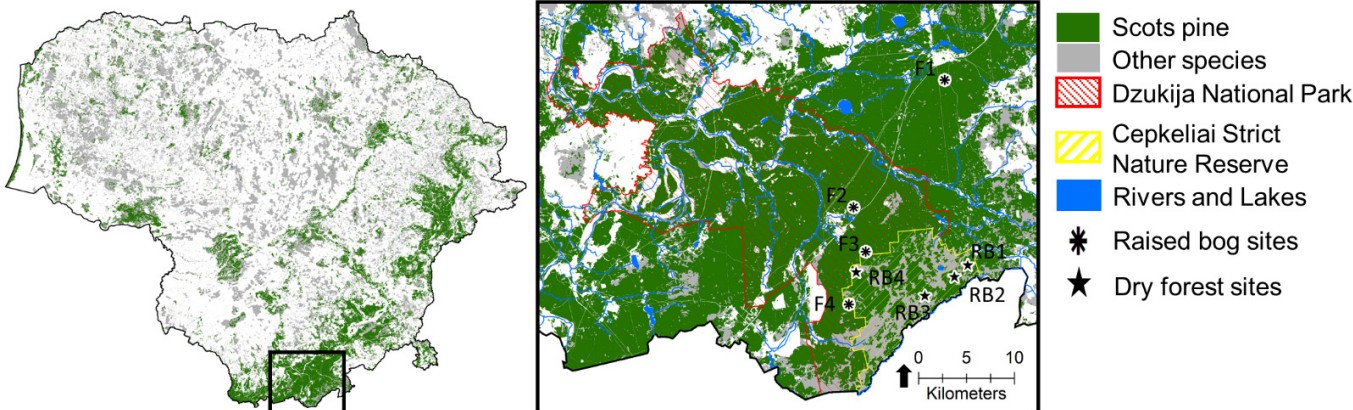

**Figure 3.** Map of Lithuania's Scots pine forest cover (**left**) and the site locations within the Dzūkija landscape for the dendrochronology study of the fire history and climate (**right**).

We applied the ASIO model to Lithuania's forests to identify the occurrence of different fire frequencies across the landscape. First, we used the regional forest site type classification, based on soil moisture and fertility, as developed by Vaičys [56], and the associated vegetation, as developed by Karazija [45], to categorize Lithuania's forests into the ASIO forest fire frequency model [16,52] (Table 1). Second, we appended our ASIO classification model to a copy of the Lithuanian Forest Services' (2017) state forest stand cadastral GIS database. Third, we mapped and analyzed the dominant forest types for each of the ASIO fire frequency categories using the GIS database.

*2.3. Dendrochronology Fire History and Climate Analysis in Dzūkija*

2.3.1. Fire History Reconstruction

Based on the mapping outcomes of the ASIO model, we chose to conduct a reconnaissance of the Scots pine forests in the Dzūkija landscape identified by the "Often" fire frequency category. Across its range, Scots pine has evolved successional strategies to cope with natural disturbances such as low-intensity fire on nutrient-poor, dry-sandy soils. These include thick, corky bark to protect the vascular cambium from fire injury; light-wind-disseminated seeds, and deep taproots that provide the strong ability to grow on infertile, dry soils [57]. However, as a truly bi-modal species, they can also survive on poor, wet sites such as peatlands [4,58], which occasionally dry up during prolonged hot weather conditions and then become fire prone. Within the Dzūkija landscape, we selected eight locations to collect fire scar samples; four were in dry pine forest stands of the "Often" category of the ASIO model (F1-4), and four were located within the Čepkeliai Strict Nature Reserve of the "Seldom" category from the ASIO model (RB1-4) (Figure 3). The basis for the selection of two unique landcover types (i.e., dry pine forest vs. peatland) was to compare their local fire histories, which should be vastly different under seemingly natural conditions, as well as to test the predicted results of the ASIO model.

Furthermore, selection of the four sampling sites within the dry pine forest was based on the local forest management stand unit and DNP records as well as consultation with local foresters and DNP staff to identify recently harvested stands (within the last 1–5 years) that were >110 years of age with possible signs of fire injury on the residual tree stumps. We visited each of the four dry pine forest stands during April–May 2019 and inspected remnant stumps for visible fire scars. The opportunistic method for collecting samples was designed to first search the entire cutover area and flag all intact stumps exhibiting

visible fire scars without advanced decay. To minimize any shortcoming in sampling, we collected and used all possible samples to construct a chronology of signature years and fire events [8,59–61]. Using chainsaws, we then cut and collected a 5–10 cm thick cookie (transverse or cross-cut sample) of each selected Scots pine stump within the sampled stands (n = 92). The mean size of each sampling site was 2 ha. The number of fire-scarred stumps varied across the stands visited, based on the visibility of cat faces, fire scar evidence, and decay condition of the remnant stumps. All samples were cut from the stumps within 30 cm of ground level [62].

**Table 1.** The ASIO concept fire frequency model [16,52], fitted to the site type, soil, and vegetation characteristics of Lithuania's hemi-boreal forest [45,56].

| Fire Frequency ASIO Model | LT Site Type | Soil Fertility | Soil Moisture | Dominant Forest Type | Field Layer | Secondary Forest Type |
|---|---|---|---|---|---|---|
| **Absent** | Nf | Very rich | Mesic | Oak | *Aegopodiosa* | Ash, Aspen, Grey alder, Hornbeam, Linden, Norway spruce |
| | Lf | Very rich | Moist | Ash, Oak | *Aegopodiosa, Carico-mixtoherbosa* | Aspen, Birch, Linden, Grey and Black alder |
| | Uc | Semi-poor | Wet | Birch | *Calamagrostidosa* | Black alder, Birch, Norway spruce, Aspen |
| | Ud | Rich | Wet | Black alder | *Filipendulo-mixtoherbosa* | Norway spruce, Ash, Birch, Grey alder |
| | Uf | Very rich | Wet | Black alder | *Urticosa* | Ash, Birch, Norway spruce |
| | Pc | Semi-poor | Very wet | Birch | *Caricosa* | Black alder, Scots pine, Norway spruce |
| | Pd | Rich | Very wet | Black alder | *Carico-iridosa* | Birch |
| **Seldom** | Sd | Rich | Dry | Oak | *Hepatico-oxalidosa* | Norway spruce, Birch, Aspen, Hornbeam, Linden, Grey alder, Scots Pine, Ash |
| | Nd | Rich | Mesic | Oak | *Hepatico-oxalidosa* | Norway spruce, Birch, Aspen, Hornbeam, Linden, Grey alder, Scots Pine, Ash |
| | Lc | Semi-rich | Moist | Spruce | *Myrtillo-oxalidosa* | Birch, Aspen, Scots pine, Oak |
| | Ld | Rich | Moist | Ash, Norway spruce | *Carico-mixtoherbosa, Oxalido-nemorosa,* | Birch, Oak, Linden, Aspen, Grey and Black alder |
| | Ua | Very poor | Wet | Scots pine | *Myrtillo-sphagnosa* | - |
| | Ub | Poor | Wet | Scots pine | *Myrtillo-sphagnosa* | Birch |
| | Pa | Very poor | Very wet | Scots pine | *Ledo-sphagnosa* | - |
| | Pb | Poor | Very wet | Scots pine | *Carico-sphagnosa* | Birch |
| **Intermittent** | Sc | Semi-rich | Dry | Norway spruce | *Oxalidosa, Hepatico-oxalidosa* | Scots pine, Birch, Aspen Oak |
| | Nc | Semi-rich | Mesic | Norway spruce | *Oxalidosa, Hepatico-oxalidosa* | Scots pine, Birch, Aspen Oak |
| | Lb | Poor | Moist | Scots pine | *Myrtillosa* | Norway spruce, Birch, Aspen |
| | Nb | Poor | Mesic | Scots pine | *Vaccinio-myrtillosa* | Norway spruce, Birch, Aspen |
| **Often** | Nbl | Poor | Mesic | Scots pine | *Vaccinio-myrtillosa* | Norway spruce, Birch, Aspen |
| | Na | Very poor | Mesic | Scots pine | *Cladoniosa, Vacciniosa* | - |
| | Sa | Very poor | Dry | Scots pine | *Cladoniosa, Vacciniosa* | - |
| | Sb | Poor | Dry | Scots pine | *Vaccinio-myrtillosa* | Norway spruce, Birch, Aspen |

A collection of 40 Scots pine samples within the CSNR peatland was undertaken under permit Nr. S-490 (9.2), granted by the Dzūkija National Park and Čepkeliai State Nature Reserve Directorate. At each of the four study site locations (Figure 3), we randomly collected 10 samples each (n = 40) from living, standing trees with visible fire scars at a height of <30 cm above ground. The mean size of each sampling site was 2 ha. These sites all had recent fire events recorded in the local forest management records to corroborate and cross-date the samples.

Following the field collection of the Scots pine transverse samples (n = 132), we moved them to the VDU tree-ring lab in Kaunas for processing. We polished the freshly cut side of each cookie sample by first using an electric planer followed by progressive sanding with 120, 220, 320, 400, 600, and 800 grit sandpaper to achieve a high-quality finish that would enable visual and graphical cross-dating of the fire scars using skeleton plotting [63] and signature year identification, such as drought or fire years, using the List method [64,65].

As reconstructions of fire history using tree-ring data is one method that can provide valuable information on the reference conditions of fire regimes [8,60], we developed both site and regional fire histories based on the frequency of occurrence of fire scars. First, we identified both the establishment year and the harvest year of each of the transverse samples using the Lithuania forest stand database. Due to updating issues within the forest stand database, which can take 10 years for records to be recorded, we also crosschecked the stand felling dates of each sampled location with a local forester and a DNP officer. Second, using a Swift microscope with 20× magnification, we established each tree's dendrochronological record from pith to bark and determined the age of each cookie sample using interannual ring-width patterns. Subsequently, for each sample, we established the fire scar dates using skeleton plots and signature years common to the sites and samples (Figure 4). It should be noted that all samples had bark and all but two samples (RB2_15 and RB3_21) had a pith. Each cookie sample was dated using a minimum of 2 transects to account for possible missing or locally absent rings and was verified by 2 trained dendrochronologists to ensure correct dating across samples.

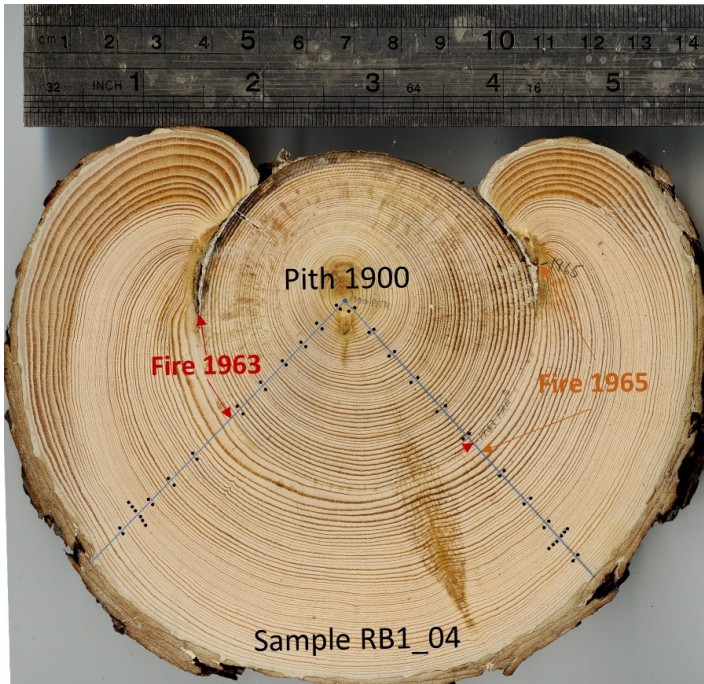

**Figure 4.** An analyzed 119-year-old Scots pine sample (RB1_04) from Čepkeliai Strict Nature Reserve peatland with visible fire scars that have been dated and used in the fire history analysis.

The Fire History Analysis and Exploration System (FHAES) [66,67] is a software tool used to analyze forest fire regimes by utilizing data recorded in tree rings to provide annually resolved dates for past fires across a range of forest types [60,68]. The software contains a built-in set of analyses that return the following key fire outputs that were used in this study: the mean fire interval, the median fire interval, the standard deviation, the minimum fire interval, the maximum fire interval, the Weibull mean, the Weibull median, and the Weibull standard deviation. Therefore, we used FHAES to analyze the past fire history using fire scars dated in the annual growth rings of 92 samples from the 4 dry pine forest stands in DNP and 40 samples from the 4 peatland sites. Following the methodological

description of Lafon et al. [60], we analyzed parameter values for the following variables: from pith date (or earliest available year) to harvest (end) date, which included the interval after the last recorded fire. Within FHAES, the alpha level was set to 0.125, the percent of trees scarred was set at 10, and the minimum number of samples was set at 3.

First, we applied individual site and composite fire archive analyses for both the dry pine forest and the peatland sites. Second, we performed a time series fire frequency analysis for pre- and post-1950, the date of effective forest fire control. The selection of 1950 as a time-period split was based on several factors; (i) intensively managed forestry practices with efficient fire protection and suppression capacities commenced in 1950 [32], (ii) it considers the national upheavals of WWI and WWII prior to 1950, and (iii), it accounts for the industrial intensification of both Lithuania's agriculture and forestry sectors under occupancy by the USSR [69–71].

### 2.3.2. Climate Variation Analysis
Precipitation Prior to Key Fire Events

We used the Superposed Epoch Analysis (SEA) Module in FHAES to assess the preconditioning of fire occurrence due to precipitation [60,68]. First, we created a yearly mean precipitation index (YMPI) time series dataset extending from 1887 to 2019, using local data [72]. This was created by dividing the yearly rainfall by a 5-year moving average precipitation (t-4current plus current year) to identify fluctuations [64]. Second, using the composite fire history reconstruction results as the key event years, we analyzed both the dry pine fire sites and the peatland sites of Dzūkija pre- and post-1950. We applied 5000 non-parametric bootstrapped simulations and assessed the statistical significance at 95% confidence of difference from the YMPI for the key composite fire years, as well as for six years prior to fire event years [60,68].

Growth–Climate Correlation Analysis

In addition to the sampling of fire scar cookies, we performed a growth–climate correlation analysis for both the dry and the peatland forest types, respectively [60,64]. We collected 44 healthy live tree core samples from the dry pine forest type sites and 40 healthy live tree core samples from the peatland pine forest type sites. Each tree was of the I Kraft class [73], and the samples were taken from breast height and measured in a south–north direction. Samples were air-dried and subsequently polished so they could be measured and statistically cross-dated using standard dendrochronological methods [64].

The analysis of the samples was performed using the tree-ring measurement station LINTAB 6. The measurements were conducted from bark to pith with the microscope set to x16 times magnification and the measurement unit of 1/100 mm. After all the samples were measured, the interannual tree-ring series were compared to existing local Lithuanian Scots pine chronologies and cross-dated using TSAP-Win software. This procedure allowed us to find any measurement errors or missing tree rings in the series. Any series that did not match were re-measured and checked again for errors. If the error persisted, the core sample was removed from later analysis. During the re-measurements and corrections, we removed 2 core samples from the dry pine forest sites and used a total of 42 tree-ring series from the dry pine sites. For the peatland sites, we found no errors.

Following Rinn [74], we applied trend elimination and calculated the ring width index (RWI) for both sites. The cross-dated series were detrended using a moving average method, and the trend function was calculated individually for every tree-ring series [64]. The chosen bandwidth length was 5 years. Furthermore, a dendroecological analysis was conducted to correlate the ring width index (RWI) for each forest site type with seasonal climatic factors to elucidate the levels of climatic influence on both fire periodicity and residual tree growth. Seasonal precipitation and temperature values for Dzūkija [72] were correlated with the RWI values of the current year (t), as well as the subsequent years, (t + 1), (t + 2), and (t + 3), to discern any relationships with tree growth through time [75].

## 3. Results

### 3.1. ASIO Fire Frequency Model Applied to Lithuania

Applying the ASIO model based on soil characteristics and vegetation revealed that all the four potential relative fire frequency categories are found in Lithuania (Figure 5, top left). Forests belonging to the "Seldom" fire frequency type were dominant (41%), followed by the "Often" type (21%), the "Intermittent" type (20%) and lastly, the "Absent" type (18%) (Figure 5, bottom left). Further analysis of the dominant forest types within the ASIO fire frequency model showed that birch (37%) and alder forests (34%) were the main forest types within the Absent category. Within the Seldom fire frequency categories, spruce (28%) and birch (29%) were the main dominant forest types, whereas within the Intermittent fire frequency, Scots pine (41%) and spruce (29%) were the main dominant forest types. Finally, within the Often fire frequency category, Scots pine was the dominant forest type, comprising 92% (Figure 5, bottom right).

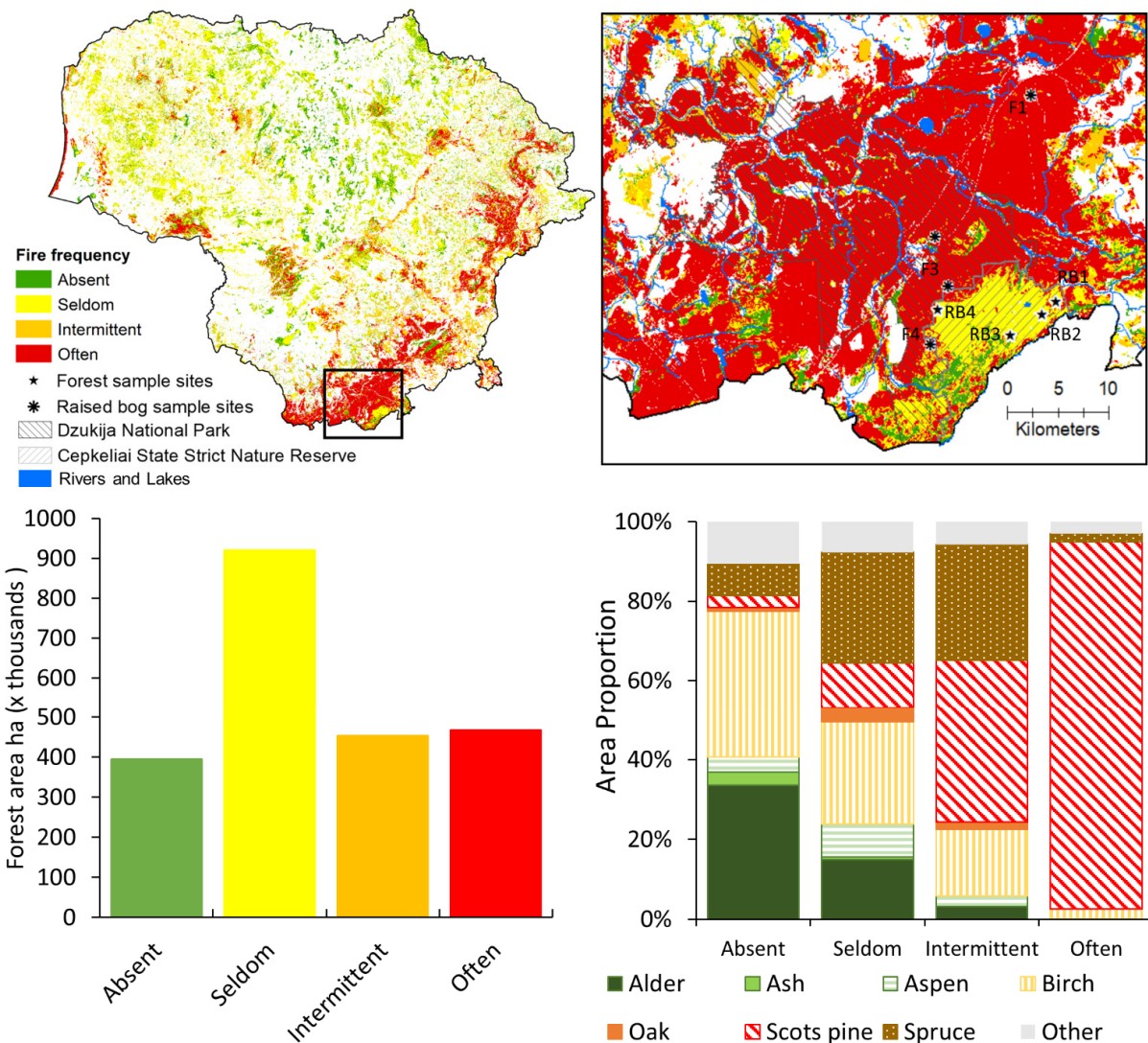

**Figure 5.** Results of the ASIO model applied to Lithuania's hemi-boreal forests. The map of Lithuania (**top left**) shows the distribution of the ASIO forest fire frequency model. The **bottom left** graph shows the area of forest within each fire frequency categories of the ASIO model. The **bottom right** graph shows the area proportions of the dominant forest types for each of the four ASIO fire frequency types. **Top right** is a map of the Dzūkija Scots pine forest landscape in southern Lithuania where the dendrochronology study was conducted based on the output of the predictive ASIO fire frequency model.

### 3.2. Fire History Reconstruction in Dzūkija

Based on the results of the ASIO fire frequency model, we collected a total of 132 fire-scarred tree samples from the Dzūkija landscape in southern Lithuania (92 from dry Scots pine forests with the Often fire frequency occurrence in the DNP and 40 from the CSNR peatland with Seldom fire frequency occurrence for the CSNR) (Figure 5, top right).

In total, the samples contained 455 fire scars dated from 1742 to 2019. The DNP dry forest sites contained 334 fire scars between 1819 and 2019 (Figure 6) and the CSNR peatland sites contained 121 fire scars between 1742 and 2019 (Figure 7). From the 8 sample sites, we identified a total of 213 unique fire events, 182 for the DNP and 31 for the CSNR. The FHAES analysis results showed the mean fire intervals were 4.3 years (SD 6.21) for the DNP's dry Scots pine forest and 8.7 years (SD 8.4) for the CSNR's peatland forest. The Weibull median fire interval was 2.7 (SD 4.75) and 6.3 years (SD 8.17), respectively (Table 2).

**Table 2.** Basic fire history statistics for Scots pine forests of Dzūkija (F) and Čepkeliai Strict Nature Reserve peatland (RB) in Lithuania.

| Composite Parameters | F1 | F2 | F3 | F4 | All F | RB1 | RB2 | RB3 | RB4 | All RB |
|---|---|---|---|---|---|---|---|---|---|---|
| Total intervals | 33 | 23 | 30 | 23 | 43 | 6 | 11 | 6 | 8 | 12 |
| Mean fire interval | 4.67 | 5 | 4.87 | 5.87 | 4.33 | 17.33 | 9.45 | 9 | 13 | 8.67 |
| Median fire interval | 4 | 3 | 2 | 4 | 2 | 18.5 | 4 | 8 | 10 | 4.5 |
| Standard deviation | 3.95 | 5.48 | 5.7 | 6.98 | 6.21 | 12.18 | 8.94 | 4.73 | 7.15 | 8.4 |
| Minimum fire interval | 1 | 1 | 1 | 1 | 1 | 2 | 1 | 3 | 6 | 1 |
| Maximum fire interval | 21 | 26 | 25 | 25 | 34 | 33 | 25 | 17 | 27 | 27 |
| Weibull mean | 4.70 | 5.04 | 4.86 | 5.85 | 4.27 | 17.2 | 9.45 | 9.02 | 13.09 | 8.68 |
| Weibull median | 3.91 | 3.79 | 3.33 | 3.9 | 2.7 | 14.6 | 6.5 | 8.64 | 12.41 | 6.29 |
| Weibull standard deviation | 3.53 | 4.49 | 4.93 | 6.14 | 4.75 | 12.32 | 9.56 | 4.29 | 6.56 | 8.17 |

Our 277-year disturbance chronology (mean tree age 130 years) of Scots pine clearly revealed that there were different phases of fire frequency. Within both the DNP dry forest and the CSNR peatland forests, we identified two principal time periods (pre- and post-1950) (Figures 6 and 7). The time series analysis results showed that the pre-1950 Weibull median fire interval was 2.22 (SD 2.96) for the dry pine forest sites and 6.2 (SD 4.4) for the peatland sites (Table 3). In comparison, the post-1950 fire scar analysis showed an increased Weibull median fire interval of 7.2 (SD 4.61) for the dry pine forest sites, whereas the Weibull median fire interval for the peatland site showed a small decrease to 5.15 (SD 8.62) (Table 3).

**Table 3.** Time series for the fire history statistics for Scots pine forests of Dzūkija National Park (DNP (F)) and Čepkeliai Strict Nature Reserve (CSNR(RB)) applied to two time periods (pre- and post-1950).

| Composite Parameters | Pre-1950 | | Post-1950 | |
|---|---|---|---|---|
| | DNP (F) | CSNR (RB) | DNP (F) | CSNR (RB) |
| Total intervals | 38 | 5 | 5 | 7 |
| Mean fire interval | 3.08 | 7 | 8 | 8 |
| Median fire interval | 1.50 | 6 | 8 | 3 |
| Standard deviation | 3.76 | 4.64 | 4.74 | 10 |
| Minimum fire interval | 1 | 1 | 1 | 2 |
| Maximum fire interval | 20 | 13 | 14 | 27 |
| Weibull mean | 3.09 | 6.95 | 7.88 | 7.95 |
| Weibull median | 2.22 | 6.19 | 7.2 | 5.15 |
| Weibull standard deviation | 2.96 | 4.4 | 4.61 | 8.62 |

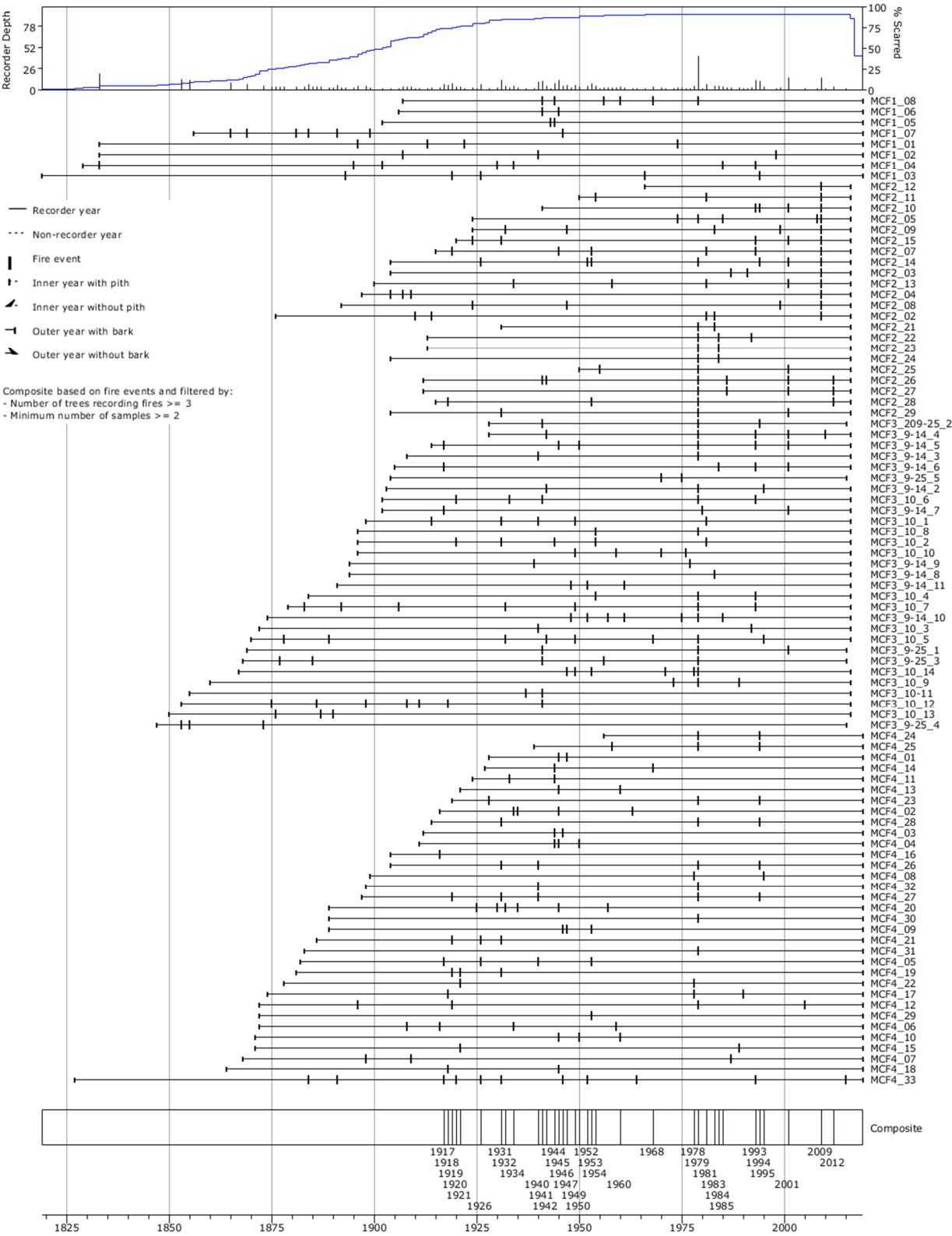

**Figure 6.** Master fire history graph created for the dry Scots pine forests of Dzūkija.

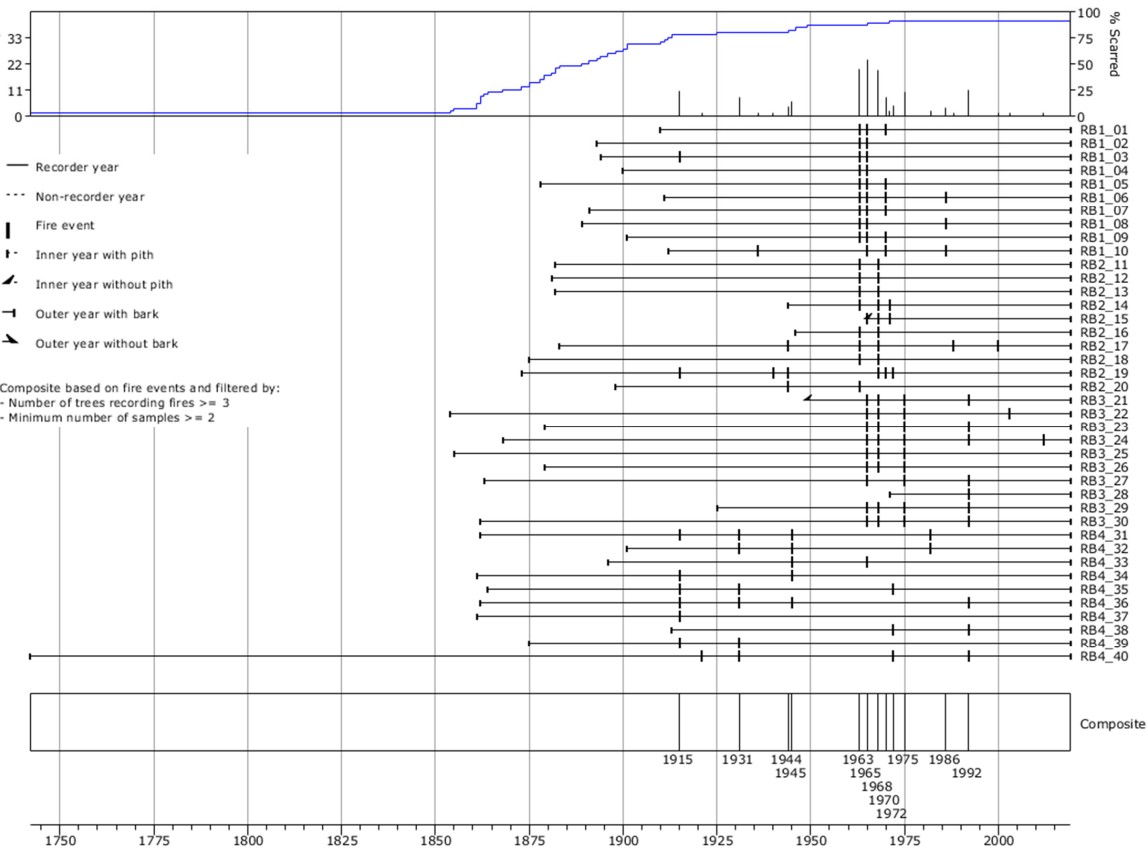

**Figure 7.** Master fire history graph created for the Čepkeliai Strict Nature Reserve's peatland Scots pine forests in Dzūkija.

### 3.3. Dendrochronology

#### 3.3.1. Precipitation Prior to Key Fire Events

The superposed epoch analysis showed fluctuations in precipitation from the yearly mean precipitation index prior to the key fire events. The pre-1950 results show both above and below average precipitation prior to the key fire years, with the dry pine sites ranging from −0.049 to 0.055 (Figure 8A) and the peatland site ranging from −0.053 to 0.098 (Figure 8B) over the seven-year period, respectively. The post-1950 results show an increased precipitation effect for the dry pine sites with a significant precipitation increase (0.127) 6 years prior to the fire events, followed by a significant decrease in rainfall (−0.127) two years prior to the fire events (Figure 8C). In contrast, the post-1950 analysis for the peatland site showed non-significant fluxes prior to the fire events (−0.075 to 0.117) (Figure 8D).

#### 3.3.2. Growth–Climate Correlation Analysis

Master chronologies for both the DNP dry Scots pine forests and the CSNR peatland Scots pine forests were calculated for the period of 1868–2018 (Figure 9 (Bottom)). Our analysis did not reveal a significant correlation between the DNP and the CSNR ring width indices ($R^2$ = 0.020, critical value 0.195). Furthermore, we did not find any significant differences between fire and non-fire years in the mean seasonal temperature (F = 0.47; $p < 0.93$) and seasonal precipitation (F = 0.52; $p < 0.90$). However, the mean growth during non-fire years was significantly higher, (1.02 mm) compared to the years with forest fires (0.993 mm (F = 18.01; $p < 0.00002$)), and current-year autumn precipitation recorded a significant correlation with the DNP master chronology ($R^2$ = 0.204, critical value 0.195). The correlation analysis for the CSNR master chronology indicated a significant relationship

with current year spring precipitation (R2 = 0.236, critical value 0.195), with no other significant relationship noted.

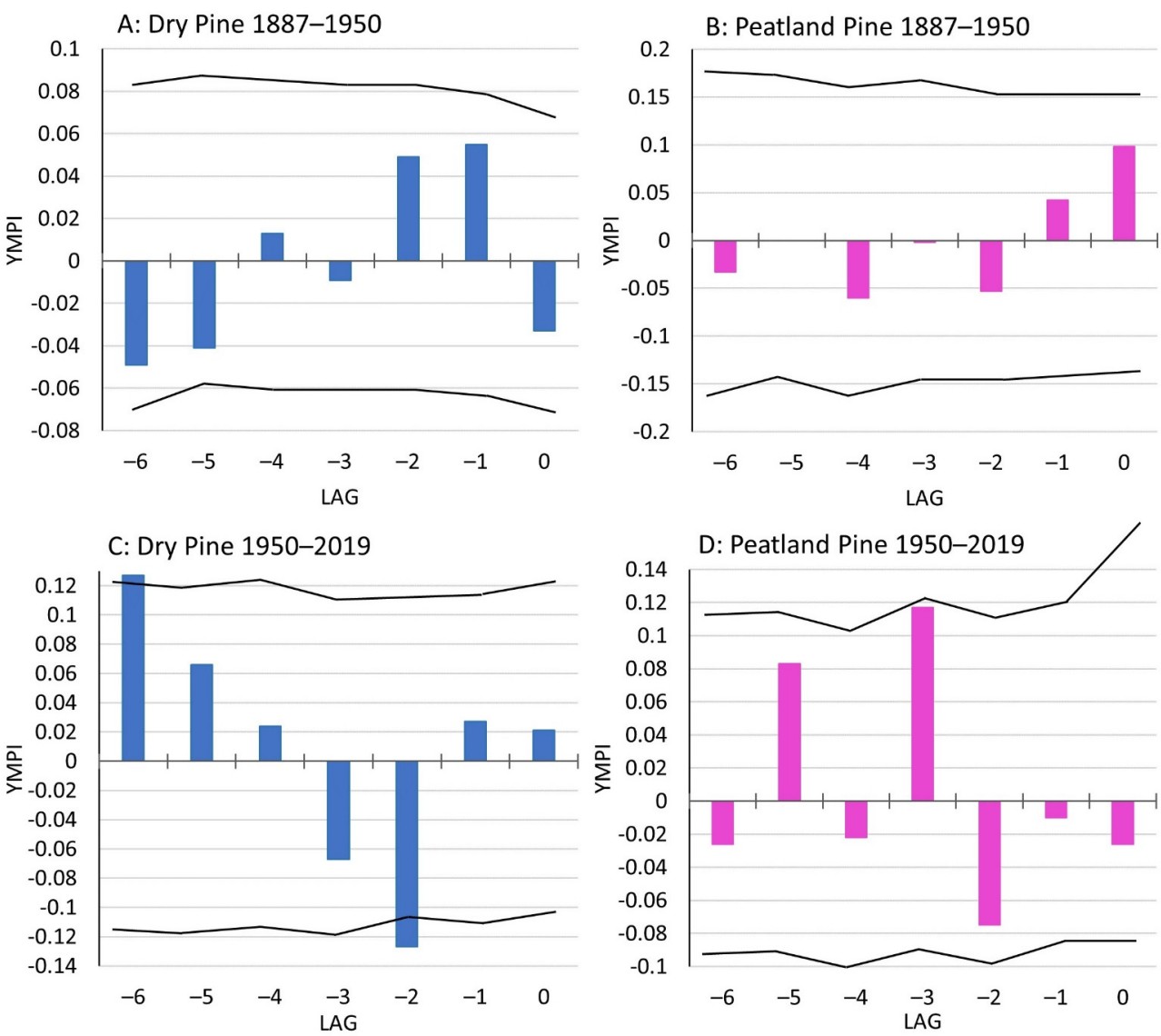

**Figure 8.** Difference in Yearly Mean Precipitation Index (YMPI) prior to the key fire events identified for two Scots pine forest site types in the Dzūkija forest landscape. The LAG year 0 represents the year of the key fire event. The black line represents the 95% confidence intervals.

In summary, the RWI for the DNP dry pine sites indicated above-average growth from 1870–1885, with below-average periodic growth in 1879 and 1889. Then, there was generally near-average growth between 1890–1975, with many concurrent fires throughout the early decades of the 20th century. High-frequency growth variation occurred for the next four decades into the 21st century. In comparison, our RWI for the peatland sites indicated average growth rates with several episodes of low-frequency variation, suggesting periodic drying and flooding of the peatland and thus supports the findings of Tamkevičiūtė, et al. [76]. On the CSNR peatland sites, fires appear to occur during the early to mid-20th century. During the last decades of the 20th century, there appear numerous high-frequency pulses with fewer fires apparent after 1988. The mean yearly precipitation and temperature indexes for the study area can be found in Figure 9 (top).

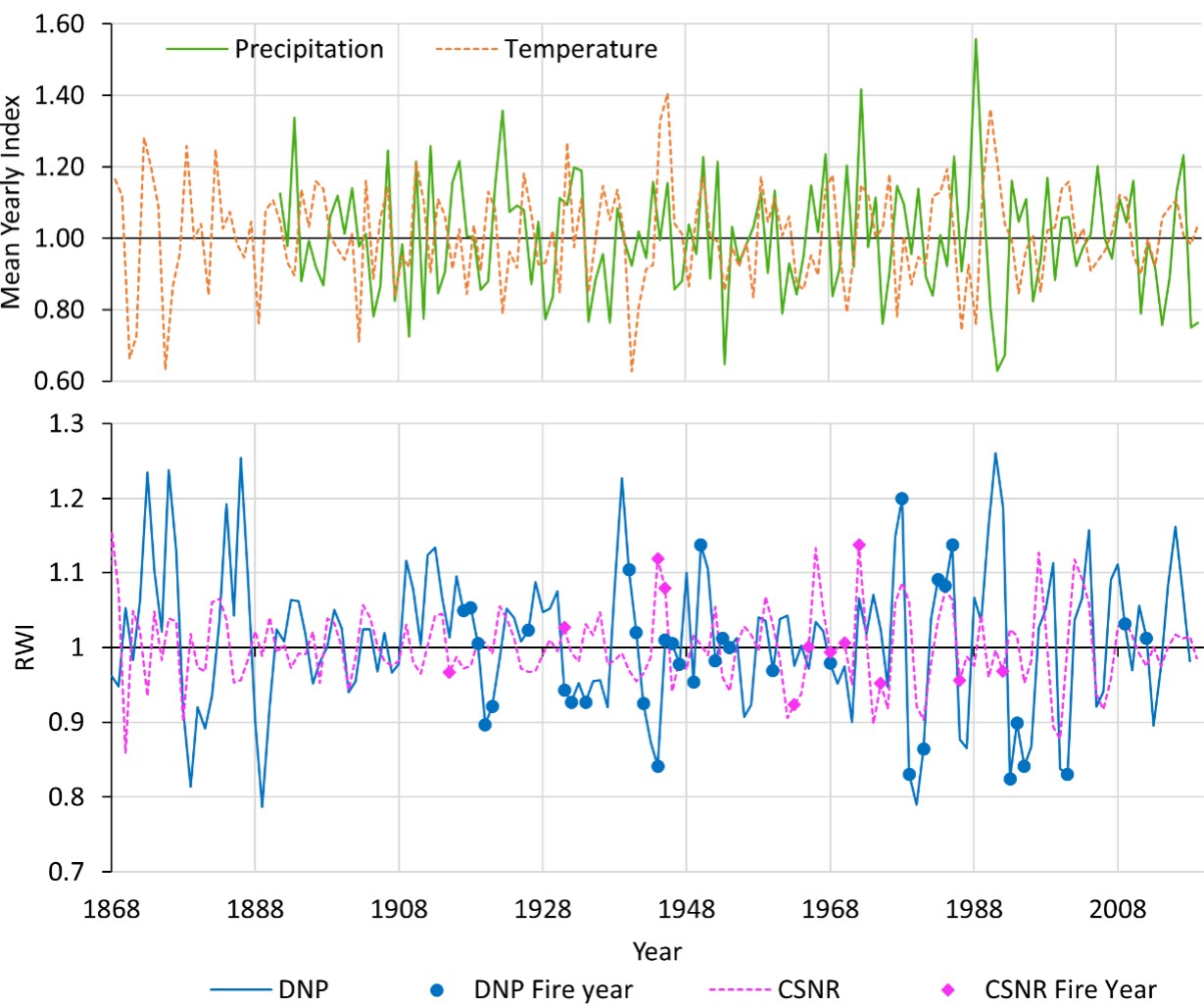

**Figure 9.** Mean Scots pine annual tree-ring width index (RWI) and key fire years for the DNP dry Scots pine forest and the CSNR peatland of the Dzūkija forest landscape (**bottom**). Yearly mean indexes showing the fluctuation in precipitation and temperature for the study area from 1868 to 2019 (**top**). Pre-1888 precipitation data were not available. Values below 1 indicate below average values, whereas above 1 indicates higher than average annual values.

## 4. Discussion

### 4.1. ASIO towards a Cultural Landscape

The application of the ASIO fire frequency model, based on Lithuania's regional forest typology [45,56], indicates that fire is a widespread potential natural disturbance in Lithuania's hemi-boreal forests. The distribution of the four relative fire frequencies in the ASIO model categories were evenly distributed in Lithuania, except for the Seldom category which was twice as common as the other three categories. This is expected given the dominance of mesic forest site types across the mid-region of Lithuania [45]. Furthermore, the ASIO model results showed a strong increase in fire-tolerant forest species, transitioning from the Absent category to the Often fire frequency category. For instance, Scots pine, being the most fire-tolerant and fire-adapted species in Lithuania, accounted for only 3% of the Absent fire frequency category but increased to a dominant 92% for the Often fire frequency category, whereas the greatest amounts of dominant broadleaf deciduous forests were found in the Absent and Seldom fire frequency categories. Thus, our results support the use of the ASIO fire frequency model to help understand the differences in forest fire frequencies and disturbance regimes.

However, today's Lithuanian forests generally do not represent the potential natural forests in terms of spatial extent and species composition [77]. Thus, it is highly improbable that they will reflect patterns and processes that match natural forest ecosystems. This suggests the results of the ASIO model for the current forest landscape could result in both under- and over-estimations of fire occurrence. For instance, in Lithuania, rich fertile forests have been converted into fields for agriculture, and mesic natural mixed forests have been transformed into forests intensively managed for sustained wood production. However, the spatial extent of the nutrient-poor Scots pine forests has generally remained constant, although in a very modified state (i.e., loss of structural complexity and stands of an even-age distribution).

The Dzūkija forest landscape, known for its dry, infertile, sandy soils, in south-eastern Lithuania, contained the largest area of the Often fire frequency category in Lithuania, according to the ASIO model. Thus, this area under natural fire regimes should be dominated by frequent low-intensity fires with low severity. As the minimum harvesting age of Scots pine in Lithuania is 110 years [78], we would argue that nearly all harvested stands of Scots pine in Dzūkija would produce cross-datable material for such fire history archives.

Our study does not provide any indication on the severity or areal extent of the fires, based on the small sampling areas, but assumptions could be made that all fires were of low severity. This can be based on our fire history reconstruction results and the fact that all our samples were able to survive multiple fire events given the undulating terrain and fragmented fuel matrix. This is also supported by Perera and Buse [31], who state that surface or ground fires are the most common boreal fire type in northern Europe, whereas more intensive crown fires are more common in Eastern Russia and North America. This also suggests that both the fire sizes and areas burnt are much smaller in northern Europe. Thus, the fires recorded in this study are more localized forest fires. However, the synchronous fire years recorded in our samples suggest that there were both larger extensive forest fires, as well as asynchronous fires, which are generally confined to much smaller areas. Moreover, regional differences in macroclimate, fire regimes, in terms of size, seasonality, fire intensity and severity, and fuel consumption have also been found [31,79] and thus support the differences in fire behavior and patterns.

Our fire history reconstruction confirms that forest fires have been an integral characteristic of the Dzūkija Scots pine forest landscape, at least for the past 200 years. Comparing the absolute frequency of fire occurrence in Scots pine forests (Often-40–60 years [6,16,55] and Seldom-100–150) on peatlands [6] with the fire history reconstruction results for Scots pine forests of the Dzūkija landscape reveals that the cultural occurrence of fire is much higher than that of natural forest ecosystems (e.g., 20–300 years in the boreal forest [14,31]). Our results show that the fire intervals ranged from 1–34 years for the dry pine sites and 1–27 years for the peatland sites over the studied period of 1742 to 2019. These ranges are similar to that found in the Poland's Białowieża Primeval Forest, south of Dzūkija, where the fire interval ranged from 1–49 years over a 350 year period [8], whereas to the north in Latvia's Slitere National Park, the fire intervals ranged from 45–80 years [9]. Thus, our study is an important contribution in conjunction with the two studies from adjacent Poland and Latvia, providing strong evidence that historic fire is a large component of the Baltic forest landscape.

The mean fire intervals of 4.33 and 8.67 recorded in this study suggest that there are other factors driving fire occurrence. This is also consistent with the variant of the ASIO model developed by Drakenberg [80] for southern Sweden, which was named AIK with the three classes A (Almost without fire), I (Intermittent fire), and K (Kultur—cultural in English) for landscapes with strong traditions of using fire as a management tool. Given that humans have been part of the Baltic landscape for approximately 11,000 years [70], we call for further research to develop fire frequency prediction models that include cultural aspects towards improving the understanding of ecological patterns and processes of fire within forest ecosystems.

*4.2. Cultural Influence on Fire history in Lithuania*

This and previous studies of forest fire history throughout the Baltic Sea Region show that fire is both a natural and cultural component of forest landscapes [8,9,81–83]. The reconstruction of local fire histories representing different site conditions, as applied in this study, is a key method as fire incident records are only available at a national level prior to 1980. However, there are limitations to using trees as fire archive records: (1) approximately less than 10% of trees are injured and scarred during most low-intensity surface fires [84,85]; (2) few old "witness trees" or "legacy trees" remain due to the past forest harvests [4]; and (3) the measuring and calculations of radial growth of samples from slow, compressed, incremental growth and missing years can be extremely difficult. Indeed, this was evident within the peatland samples from the Čepkeliai Strict Nature Reserve. Finally, (4) the first occurrence of a fire and tree age, once a fire scar is established at a young age subsequent fires are more effectively recorded than if a fire must produce the first scar [61]. In our records, the mean tree age of the first fire scar was 39 years (range 3–113) for the dry forest sites and 64 years (range 16–179) for the peatland forest sites. This may be why our records report limited fires in the earlier years of the study, especially for the CSNR peatland sites. Therefore, the Dzūkija forest landscape has most likely experienced more fires than is even acknowledged in this study.

Intact forest landscapes provide a reference point towards understanding the natural disturbances [21,86]. However, this overlooks the fact that humans exist and are part of the landscape. For instance, lightning is the only natural cause of fire ignition in the boreal forests of northern Europe (Estonia, Finland, Latvia, Lithuania, Norway, and Sweden), but is estimated to ignite only 7.3% of all fires [87]. In Lithuania, anthropogenic use of fire as a management tool was once widespread in times of both peace and war. Current Lithuanian records indicate that only 1% of forest fires are ignited by lightning [88]. This suggests that humans are the dominant igniters of fire in this region, which concurs with observations in the forest of the Dzūkija landscape that most of the ignitions were caused by humans. However, during a field trip within the DNP and CSNR, the local park ranger informed us that the Čepkeliai peatland was able to influence weather patterns due to its size and then proceeded to show us an elevated area at the edge of the CSNR peatland where nearly every Scots pine tree within a 100 m radius had signs of being struck by lightning but with very few apparent signs of fire extending outside this area. Thus, understanding the historical range of variability [35,89] and the effect of human influences on ecological patterns and processes of fire [12,19,90] is an important complement to the natural range of variability for understanding how forest management systems can be developed [13]. Analyzing the fire frequency across different periods of land use, cultural change, and governance, as well as political upheaval (e.g., pre- and post-1950, as in this study) illustrates the human influence on fire ignitions and subsequent control. In this study, the division of reconstructed fire history into the pre- and post-1950 eras shows that the mean pre-1950 fire interval (3.08) was more than double compared to that of post-1950 (8.00), whereas the differences in the peatland sites were 7 and 8, respectively. This suggests that the fires in the CSNR site were more climate driven, as opposed to the dry sites of the DNP sites. However, Tamkevičiūtė et al. [76] found that the responses to climate in the CSNR peatland can be delayed. Indeed Kitenberga et al. [9] also identified periods of high and low fire activity pre- and post-1950 and linked them to social political change in Latvia. This suggests that there are different eras and periods of land use and intensification, as well as political instability [19,70]. The effects of Soviet land use and control over this landscape post 1950 to established broad control measures to inventory, compartmentalize, and protect forested areas under state control are well documented and were apparent from our time series analysis.

Reviewing key cultural events of Lithuania and the Dzūkija landscape illustrates the effects of fire and can even be aligned with the key fire years found in our study (Table 4). Over the past 250 years, many cultural, political, and management events have transformed the forest landscapes. During both World War I (1914–1918) and World War II (1939–1945),

front lines and demarcation borders were established in Lithuania [91]. Our results reflect increased fire frequency during these periods, as would be expected when occupiers burned farms, structures, and the woods to root out partisans and enemy collaborators. Between these two wars, the Dzūkija landscape was annexed to Poland. However, after World War II, Lithuania regained Dzūkija and was forced into the Soviet Union, resulting in large land reforms that swept through the country after 1950. This led to industrial land management with increased production in the agricultural sector and extensive afforestation of poor sites of the Dzūkija region [69,71]. However, it has been suggested that the regional forests were less affected and better conserved through creative reporting and underestimations of residual stocking, which led to subsequent imports of timber from other parts of the Soviet Union [47]. A review of fire history records for Lithuania shows that since approximately 1950, forest fire suppression and control efforts and capacities have greatly improved (Table 4). This is also supported by our results from the pre- and post-1950 fire frequency time series analysis.

**Table 4.** A time periods outline of Lithuania's fire statistics [48,88], and key cultural events that have influenced both Lithuania's and Dzūkija's forest fire history. * Specific to the Dzūkija landscape. + Dzūkija region was under Polish administration 1922–1939, and fire records were not included in the reporting.

| Time Period | Number of Fires | Area Burnt (ha) | Mean Fire Size (ha) | Cultural Influences on Forest Ecosystems |
|---|---|---|---|---|
| 1795–1914 Imperial Russia | No data | No data | No data | Control, noble estates, and serfdom, 1859–1862 Warsaw–Saint Petersburg railway construction * 1863–1864 Lithuanian uprising Lands were marked by violence and guerrilla warfare [70] No fire control systembut fire was used for traditional land use Mosaic landscape Burning for subsistence farming (increased berry yields) Fire was used to clear forest land for agriculture Katra and Ula Rivers bifurcation (flow change (7 km)) * |
| 1914–1918 WWI | No data | No data | No data | Front line passage * 1914 Narrow railway construction (28.5 km) Marcinkonys–Čepkeliai for wood extraction * Big fire 1915 several fires totaling thousands of hectares * |
| 1918–1939 First Independence + | 2053 + | 15908 + | 10.41 + | Widespread forest planting * Private forest, no fire control Intensive fire protection by Polish state forests * |
| 1939–1945 WWII | 1052 | 15285 | 12.97 | Front line passage, war activities were common Marcinkonys ghetto established 1941–1942 * |
| 1945–1990 Soviet occupation | 3166 | 25316 | 7.27 | 1945–1953 Stalin's reign of terror and upheaval |
| | 13101 | 8343 | 0.63 | Widespread forest planting Soviet army forest stealing and illegal activities Collective farms, industrial agriculture and farming, no private land ownership Relocation of people from Dzūkija to other parts of Lithuania 1963–1975 climate effects, peak fire events * 1963–1968 large wildfires in CSNR, need to evacuate nearby villages * Human mobility increased, leading to accidents and arson Effective fire control, even-aged forest management |
| 1990–2016 2nd Independence | 13638 | 7233 | 0.51 | Forest ownership restitution 2004 EU membership Effective fire control Even-age stand management Forest harvesting doubled Increased fuel loads, high risk for fire events 99% of forest fires are from anthropogenic ignition |

Due to different reasons, official historic documentation of forest fires is lacking; for instance, official documents for the CSNR have only been recorded since 1983. Thus, historic records likely do not include all fires. This is especially noted for the era prior to 1950. Nevertheless, severe drought and fire events from 1963–1968 were also reflected in both our fire-reconstruction records and in the memories of the inhabitants of nearby villages surrounding the Čepkeliai peatland. Local people still remember the droughts and abandoning homesteads (burying their most cherished possessions) and evacuating

their villages (by horse or on foot) to escape the significant fire years of 1963 and 1965. More recently, rural areas in Lithuania have experienced the highest emigration rate in the European Union [92] which has led to large-scale land abandonment of agriculture and subsequent expansion of grass with sporadic Scots pine regeneration, which is particularly prone to anthropogenic fires as there are no management practices. This is certainly a problem within the Dzūkija landscape and presents a future fire risk [19,93].

*4.3. Dendrochronology, Climate Change and Fire Risk*

The superposed epoch analysis of the pre- and post-1950 peatland sites showed that the fire events were not significantly influenced by short-term precipitation events. A similar result was recorded for the pre-1950 dry pine sites. However, for the post-1950 dry pine sites, a significantly higher precipitation was recorded 5 years prior to the fire events. This suggests that there was a preconditioning of fire occurrence through an increase in fine fuels produced in response to increased precipitation [68,94]. Following this period, a drying out effect with a significant precipitation decrease, just 2 years prior to the fire event was evident. Thus, the effects of precipitation appear to be a precursor of fire occurrence for the dry Scots pine sites [95,96].

Although the climate–growth analysis did not show any significant differences between fire and non-fire years for the dry pine sites, there is evidence that fire occurrence is also related to climatic events [97,98]. The main causes for the reduced growth of tree rings (signature years) are climatic events such as droughts, winter temperatures, and spring frosts [99,100]. Studies on Scots pine with a master chronology from Lithuania found multiple years with significantly reduced growth [99,101]. In our study, we identified multiple signature years on the dry Scots pine sites (1879, **1889**, 1920, 1943–1944, **1979–1981**, 1986–198**7**, 1993–199**6**, 2000–2001, and 2013) with reduced growth, of which some match (indicated in bold).

Indeed, many of the years with reduced tree-ring growth, for both our study and that of Vitas [99], coincide with the composite fire years produced by FHAES and presented above. For instance, from 1963–1977 a prominent period of drought was recorded in Lithuania [76,102]. Our reconstructed fire history also indicates increased fire activity for the CSNR peatland sites during this period, which are generally wet and fall within the "Seldom" type of the ASIO fire frequency model. This could be attributed to the drying up of the peatland during prolonged drought. During this 1963–1977 period, we also found increased tree-ring growth. There could be several ecological factors driving these patterns. Firstly, the fires provided nutrients in a normally nutrient-poor environment [31] that stimulates growth in combination with the drying or lowering of the water table of the peatland. However, the effect of the fire and response in ring growth is difficult to obtain as we were unable to identify the seasonal timing of the fire. For instance, a spring fire could trigger increased growth commencing in the same year, whereas the effects of a late-autumn fire may not be recorded until the following year. As for the dry Scots pine sites, we found increased fire activity from 1978–1984, which also coincides with drought. The year 1980 has been identified as a signature year in dendrochronology research for Lithuania due to suppressed tree-ring growth caused by climate variation [99,103,104].

Climate data from recent decades within our study region show increased temperature, especially in March–April and July–August, and slight increases in precipitation, without compensating for increased evapotranspiration [104,105]. Scots pine growth modelling predicts increased growth and biomass on mineral soils (dry sites) due to increased spring temperatures, but harsher growing conditions and a decrease in tree growth are predicted for the CSNR peatland due to extreme changes in the water table [105]. Variations in the hydrothermic regime of the CSNR peatland have caused the water table to fluctuate prominently [76]. Considering these factors suggests that the potential risk of forest fires for the Dzūkija landscape is increasing due to climate change. This includes increases in the frequency, intensity, and severity of natural and anthropogenic forest fires [40,106].

*4.4. Barriers and Bridges towards Emulating Fire in Scots Pine Forests*

The cultural influence through both traditional and intensive landscape management has shaped the European landscape and its ecosystems [19,90] and altered ecological patterns and processes [3,18,107–109]. The notion that clear-cut harvesting can substitute the natural processes of fire is a misconception [110,111]. Under natural conditions, Scots pine forests are governed by frequent low-intensity fires that lead to multi-cohort dynamic forests [14]. This process produces stands that contain several cohorts of trees which were established after previous successive fire disturbances and thus results in complex, multi-layered, and varied-age-structured stands with no, or a poorly developed, deciduous phase [16,112]. The supply of snags and dead wood on the ground in different stages of decay is often continuous since decay rates are low and old standing trees may fall over a long time after dying. Natural disturbances, such as fire, also leave important biological legacies (e.g., dead wood, seed bank, structure, undisturbed patches) that help ecosystem recovery.

Under current Lithuanian legislation, the burning of forest and agricultural land is strictly prohibited, which is incongruous with its cultural history and forest traditions. Modern forest management has substituted low-intensity fire with the use of clear-cut harvesting and deep mechanical scarification. This has simplified both the age profile and structural complexity of forest stands. Thus, many of the forest ecosystems that have evolved with fire are extremely vulnerable to losses of their typical ecological patterns and processes. Research shows that deep soil scarification after clear-cut harvesting causes major damage to the topsoil profile, negatively effects natural succession, and releases trapped carbon from the soil [113]. In comparison, low-intensity prescribed fire has been found to preserve the topsoil and microorganisms [108] and even help to maintain stored carbon reserves [114,115]. Other research on low-intensity prescribed burning in forest ecosystems has shown multiple benefits, including improved natural seed regeneration [116–118], bilberry, and lingonberry (*Vaccinium myrtillus* and *Vaccinium vitis-idaea*) cover and fruit production [119], increased rare saproxylic and red-listed beetles [120], spiders and capercaillie (*Tetrao urogallus*) [121], and insect and woodpecker occurrences [31]. Moreover, Marozas et al. [101], found that both species richness and ground vegetation coverage of Scots pine stands increased after low-intensity fire in eastern Lithuania. However, our study shows that fire as a process of the Dzūkija forest landscape is currently missing due to effective restrictive policy, fire control systems, and clear-cut forest management that focuses on wood production. Thus, the introduction of controlled, experimental, low-intensity and low-severity prescribed fires into the dry Scots pine forest stands of Dzūkija (i.e., in the Often category of the ASIO model) could be the next step towards emulating both natural and historic cultural forest disturbances [e.g., 110] and moving the public towards understanding and accepting the multiple benefits of fire for sustainable forest management. To help support the EU Forest Strategy [34] in reaching sustainable forest management, we suggest conservative prescribed burning trials should commence in the Dzūkija dry pine forests that target 50–100 ha per year with pre- and post-fire monitoring using adaptive forest management [122].

However, there are also many cultural barriers that need to be bridged; for example, a study on the perceptions of fire in Lithuania found that 90% of respondents were against the use of fire as a forest management tool and only 5% of the respondents agreed that fire is part of the forest landscape [93]. Furthermore, a survey by Pereira, Mierauskas and Novara [42] found that the public thinks prescribed fire should not be used as a landscape management tool. However, on a positive note, the respondents agreed that fire has a positive effect on biodiversity. This lack of acceptance of fire can be connected to the misunderstanding of the historic role of fire and how it effects the ecological patterns and processes of forest landscapes and by the perception of the direct effects of economic loss [123]. This reveals that the general public does not fully understand the role of fire in creating and maintaining today's forest landscape in Dzūkija and that educational programs

should be initiated across the national parks of Lithuania to reveal the long history of fire on these forested landscapes.

To encourage multi-functional forest landscapes that can provide broad portfolios of ecosystem services requires improved communication, education, and public awareness. Communication could focus on disseminating the benefits of fire and the results from the use of prescribed fire from already completed EU projects. Education should be encouraged to include knowledge about forest disturbance regimes, natural and cultural fire occurrence and frequency, and how suitable sites for prescribed burning of Scots pine forests can be located. Finally, public awareness and collaborative learning can be encouraged by applying a place-based, landscape-level approach that collects local knowledge which is shared by stakeholders and actors at multiple levels for broad acceptance and support [124,125].

## 5. Conclusions

This study shows that fire, as both an ecological process and due to historic events and cultural traditions, is part of the Lithuanian forest landscape. According to the ASIO model, approximately 20% of Lithuania's forests have a fire frequency belonging to the Often category, which comprises of 90% of the Scots-pine-dominated forests. The reconstruction of a local fire history in Dzūkija, southern Lithuania, suggests that the incidence of cultural fire is much higher than naturally caused ignitions in the ASIO model. Our study supports the growing evidence that ecosystem legacies can act as a driver for adaptive management towards achieving sustainable forest management goals in terms of both production and conservation of biodiversity and unique species assemblages for building resilient ecosystems in the face of climate change. Our study reveals that the examination of biological archives, such as historic fire occurrence, can be used to guide forest management, but at this time, low-intensity prescribed fire is prohibited by statute. We recommend introducing educational programs in Lithuania's national parks system to communicate the historic use, the ecological values and benefits of forest fire, as well as implementing adaptive management trials that use low-intensity prescribed burning of Scots pine stands, aimed towards close-to-nature forestry and the mimicking of natural forest disturbance regimes, and maintaining unique species and their habitats.

**Author Contributions:** Conceptualization, M.M. and C.R.; methodology, M.M., C.R., G.B. and P.A.; field work and data collection, M.M., C.R., G.K., G.B. and V.M.; formal analysis, M.M., C.R., R.P. and E.M.; writing original draft, M.M. and C.R.; editing, M.M., C.R., G.K., G.B., V.M., R.P. and P.A.; revision, M.M., C.R. and P.A.; visualization figures and tables, M.M. All authors have read and agreed to the published version of the manuscript.

**Funding:** This research was funded and supported by the LIFE-IP PAF-NATURALIT (Project number LIFE16IPE/LT/016).

**Data Availability Statement:** Not applicable.

**Acknowledgments:** We thank Mindaugas Lapelė from the Dzūkija National Park for his bright insights into the Dzūkija landscape and field assistance and the staff of Varena State Forest Enterprise for their field assistance.

**Conflicts of Interest:** The authors declare no conflict of interest. The funders had no role in the design of the study; in the collection, analyses, or interpretation of data; in the writing of the manuscript; or in the decision to publish the results.

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
