# Peer review of "Fire Occurrence in Hemi-Boreal Forests: Exploring Natural and Cultural Scots Pine Fire Regimes Using Dendrochronology in Lithuania"

_land, doi:10.3390/land11020260_

Round 1
Reviewer 1 Report
This manuscript describes the past fire histories of Lithuanian forests and the future of these forests. It is a very well-written manuscript. I enjoyed reading the history of the forests, but some of the history could probably be shortened. I appreciate the maps and figures, but have a few suggestions below to clarify them. I believe that this manuscript should be accepted after minor revisions. Lines 163-170: This seems repetitive according to Figure 1. I think this section could be removed or moved to the relevant sections. Lines 182-183. These are partial sentences. Lines 293-295: Were these tree core samples? Figure 8: Can you include standard deviations in these graphs? Figure 9: Consider moving the DNP and CSNR to separate stacked graphs for a better comparison between the two areas. Also, I think it would be helpful to include precipitation and temperatures along with these graphs or in a separate figure. Line 421: I know that “Often” is from the ASIO model, but I don’t think it needs to capitalized here. Line 468: You are missing a parenthesis in this sentence. Lines 515-519: Rephrase this sentence as a personal communication from M. Lapelė.Author Response
Dear Reviewer 1.
Thank you for your encouraging review. We value all of your comments and suggestions and have tried to incorporate all the best we could. In most instance we agree with you and in some we disagree, but try to provide and explanation why we disagree. Below you will find the detailed response to all you queries.
Warm regards from the Authors
Reviewer 2.
Lines 163-170: This seems repetitive according to Figure 1. I think this section could be removed or moved to the relevant sections.
Authors response: Thank you for your suggestion. We think the text provide a clear definition of what we do and provides clarification for figure 1. Therefore, we have decided to leave this text as is. We hope you understand.
Lines 182-183. These are partial sentences.
Authors response: good point, edited. It now reads “These are Absent with permanently wet soils where natural fire occurrence is rare (ca. 300-year fire cycles) [53, 54], Seldom having low fire frequency on moist sites (ca. 100 - 150 year fire cycles) [6], Intermediate fire frequency on mesic sites (ca. 80 - 100 year fire cycles) [6] and Often fire frequencies on dry and poor forests occur approximately every 40 - 60 years MFI [6, 55].”
Lines 293-295: Were these tree core samples?
Authors response: Yes, this were edited to read “live trees core samples”
Figure 8: Can you include standard deviations in these graphs?
Authors response: Thanks for your question. We have followed the guidelines of Grissino-Mayer2001 who developed the software and therefore will would like to keep the figure as it is exported out of the software. This also follow other papers such as Aldrich, S. R., Lafon, C. W., Grissino‐Mayer, H. D., & DeWeese, G. G. (2014). Fire history and its relations with land use and climate over three centuries in the central Appalachian Mountains, USA. Journal of Biogeography, 41(11), 2093-2104. And Lafon et al., 2017 referenced in this manuscript.
Figure 9: Consider moving the DNP and CSNR to separate stacked graphs for a better comparison between the two areas. Also, I think it would be helpful to include precipitation and temperatures along with these graphs or in a separate figure.
Authors response: Thanks for the suggestion. We tried the stacked appearance, but this was undesirable as it does not follow the typical ring width grow index figures used in many fire history papers. Please note it is de-trended data and shows the yearly variation form the mean moving window values. However, based on your comments we added and extra figure to Figure 9 (the mean yearly precipitation and temperature indexes) and edited the figure text. It now reads “ Figure 9. Mean Scots pine annual tree ring width index (RWI) and key fire years for the DNP dry Scots pine forest and the CSNR peatland of the Dzukija forest landscape (bottom). Yearly mean Indexes showing the fluctuation in precipitation and temperature for the study area from 1868 to 2019 (Top). Precipitation data pre 1888 was not available. Values below 1 indicates below average values, whereas above 1 indicate higher than average annual values.”
Line 421: I know that “Often” is from the ASIO model, but I don’t think it needs to capitalized here.
Authors response: Based on comments form the Academic editor we rewrote this sentence. It now reads “Application of the ASIO fire frequency model, based on Lithuania’s regional forest typology [45, 56], indicates that fire was a widespread natural disturbance in Lithuania’s hemi-boreal forests.”
Line 468: You are missing a parenthesis in this sentence.
Authors response: nice find, added.
Lines 515-519: Rephrase this sentence as a personal communication from M. Lapelė.
Authors response: This sentence was edited as requested.
Thanks again!!!
Reviewer 2 Report
The topic of the article is extremely relevant, since huge areas of forests around the world are damaged by fires of varying intensity. In addition, due to global climate warming, an even greater spread of wildfires is predicted. The relevance of the research is well justified in the article. The goals and objectives are also clearly formulated. The objects of research and methods are described in detail. The state of the problem is described. At the same time, a sufficient number of literary sources have been analyzed. The methods are chosen correctly and correspond to the tasks set.The scientific novelty consists in modelling the natural forest fire regimes based on soil fertility and moisture and vegetation, performing a localised dendrochronology study of fire history and climate for the most fire prone area within Lithuania. This study used a nested regional and local study area approach, and used the ASIO-model and dendrochronology to calculate different fire frequencies based on local and regional site conditions. It should also be noted the excellent visualization of the results. 9 figures and 4 tables are given in paper. All figures are of excellent quality and do not duplicate tables in content. The study supports the growing evidence that managing ecosystem legacies can act as a driver for adaptive management to achieve forest management goals in terms of both production and conservation of biodiversity and unique species assemblages for building resilient ecosystems in the face of climate change. The conclusions are reasonable and follow from the text of the paper. In general, the paper is read with interest, it is completely understandable and will undoubtedly be useful for forest management purposes.
I have a wish: to clarify the forest typology used, to indicate references to the authors of regional typological classification. It is important because there are many approaches to classifying forests. And the conclusions and their reliability largely depend on the classification adopted for use.
Author Response
Dear Reviewer 2,
Thank you for the very nice review, We are very happy that you consider our manuscript worthy for publication. As to you small wish please see below.
R2. I have a wish: to clarify the forest typology used, to indicate references to the authors of regional typological classification. It is important because there are many approaches to classifying forests. And the conclusions and their reliability largely depend on the classification adopted for use.
Authors response: We have edited the text line 189 - 191 to be clearer. It now reads “First, we used the regional forest site type classification, based on soil moisture and fertility, as developed by Vaičys [56], and associated vegetation as developed by Karazija [45] to categorize Lithuania’s forests into the ASIO forest fire frequency model [16, 52] (Table 1).”
Also added references in the first sentence of the Discussion.
Thank you form the Authors